# IMPROVING MULTI-MANIFOLD GANS WITH A LEARNED NOISE PRIOR

## ABSTRACT

Generative adversarial networks (GANs) learn to map samples from a noise distribution to a chosen data distribution. Recent work has demonstrated that GANs are consequently sensitive to, and limited by, the shape of the noise distribution. For example, a single generator struggles to map continuous noise (e.g. a uniform distribution) to discontinuous output (e.g. separate Gaussians) or complex output (e.g. intersecting parabolas). We address this problem by learning to generate from multiple models such that the generator's output is actually the combination of several distinct networks. We contribute a novel formulation of multi-generator models where we learn a prior over the generators conditioned on the noise, parameterized by a neural network. Thus, this network not only learns the optimal rate to sample from each generator but also optimally shapes the noise received by each generator. The resulting Noise Prior GAN (NPGAN) achieves expressivity and flexibility that surpasses both single generator models and previous multi-generator models.

## 1 INTRODUCTION

Learning generative models of high-dimensional data is of perpetual interest, as its wide suite of applications include synthesizing conversations, creating artwork, or designing biological agents (Bollepalli et al., 2017; Tan et al., 2017; Blaschke et al., 2018). Deep models, especially generative adversarial networks (GANs), have significantly improved the state of the art at modeling these complex distributions, thus encouraging further research (Goodfellow et al., 2014). Whether implicitly or explicitly, works that use GANs make a crucial modeling decision known as the *manifold assumption* (Zhu et al., 2016; Schlegl et al., 2017; Reed et al., 2016). This is the assumption that high-dimensional data lies on a single low-dimensional manifold which smoothly varies and where local Euclidean distances in the low-dimensional space correspond to complex transformations in the high-dimensional space. While generally true in many applications, this assumption does not always hold (Khayatkhoei et al., 2018).

For example, recent work has emphasized situations where the data lies not on one single manifold, but on multiple, disconnected manifolds (Khayatkhoei et al., 2018; Gurumurthy et al., 2017; Hoang et al., 2018). In this case, GANs must attempt to learn a continuous cover of the multiple manifolds, which inevitably leads to the generation of off-manifold points which lie in between (Kelley, 2017).

The generator tries to minimize the number of these off-manifold points, and thus they are generally just a small fraction of the total generated distribution. As such, they barely affect the typical GAN evaluation measures (like Inception and FID scores for images), which measure the quality of the generated distribution as a whole. Thus, this problem is usually ignored, as other aspects are prioritized. However, in some applications, the presence of these bad outliers is more catastrophic than slight imperfections in modeling the most dense regions of the space. For example, consider the goal of an artificial agent acting indistinguishably from a human: the famous Turing Test. Incorrectly modeling sentence density by using a given sentence structure 60% of the time instead of 40% of the time is relatively harmless. However, generating a single gibberish sentence will give away the identity of the artificial agent.

Moreover, there are serious concerns about the implications this has for proofs of GAN convergence (Mescheder et al., 2018). These works address the problem of disconnected manifolds by

Figure 1: The Noise-Prior GAN (NPGAN) architecture. Unlike previous work, the NP network learns a prior over the generators conditioned on the noise distribution $z$. This allows it to both control the sampling frequency of the generators and shape the input appropriate to each one, in an end-to-end differentiable framework.

simultaneously training multiple generators and using established regularizations (Chen et al., 2016) to coax them into dividing up the space and learning separate manifolds.

Methods for getting multiple generators to generate disconnected manifolds can be divided into two categories: (i) imposing information theoretic losses to encourage output from different generators to be distinguishable (Khayatkhoei et al., 2018; Hoang et al., 2018) (ii) changing the initial noise distribution to be disconnected (Gurumurthy et al., 2017). Our approach falls into the second category. Previous efforts to change the noise distribution to handle disconnectedness has exclusively taken the form of sampling from a mixture of Gaussians rather than the typical single Gaussian (with sampling fixed and uniform over the mixture).

Our approach differs significantly from those previously. We use multiple generators as before, but instead of dividing up the noise space into factorized Gaussians and sending one to each generator, we let an additional neural network determine how best to divide up the noise space and dispatch it to each generator. This network learns a prior over the generators, conditioned on the noise space. Thus, we call our additional third network a noise-prior (NP) network. Previous methods have modeled the data with noise $z$ and generators $G_i$ as $p(G_i|z) \cdot G_i(z)$, with $p(G_i|z) = p(G_i) = p(G_j) \ \forall i, j$. We instead propose a framework to incorporate a richer $p(G_i|z)$ into the generator. This framework is entirely differentiable, allowing us to optimize the NP network along with the generators during training.

We note that with this strategy, we significantly increase the expressivity of each generator over the previous disconnected manifold models. By dividing up the space into four slices $s_i$ and sending $s_1, s_3$ to the first generator and $s_2, s_4$ to the second generator, we can generate four disconnected manifolds with just two generators. Previous work would have to devote precisely four generators to this task, with degradation in performance if fewer or more generators are chosen for the hyperparameter. Here, the prior network learns to divide the noise space appropriately for whatever number of generators is chosen, and is thus more expressive as well as more robust than previous models.

Moreover, much existing work has exclusively framed the problem as, and tailored solutions for, the disconnected manifold problem. Our approach is more generalized, addressing any misspecification between noise distribution and the target distribution. This means that our approach does not become redundant or unnecessary in the case of single complex manifolds, for example.

Our contributions can be summarized as:

1. We introduce the first multi-generator ensemble to learn a prior over the noise space, using a novel soft, differentiable loss formulation.

2. We present a multi-generator method that can learn to sample generators in proportion to the relative density of multiple manifolds.

3. We show how our model not only improves performance on disconnected manifolds, but also on complex-but-connected manifolds, which are more likely to arise in real situations.

## 2 RELATED WORK

Several previous works have included multiple generators, mixing and matching a few commonly used features. Some use completely distinct generators (Khayatkhoei et al., 2018; Arora et al., 2017), while others tie some or all of their weights (Gurumurthy et al., 2017; Hoang et al., 2018).

Most use a single parametric noise source (e.g. a single Gaussian) (Khayatkhoei et al., 2018; Hoang et al., 2018) while one uses a mixture of Gaussians (Gurumurthy et al., 2017). Most sample the generators randomly with equal probability, but one attempts to find (in a non-differentiable way) a sampling scheme to not sample from redundant generators (Khayatkhoei et al., 2018). (Ghosh et al., 2018) encourages diversity among generator outputs by introducing a classifier that tries to identify the generator a data point came from, or whether it is a real data point (reminiscent of an auxiliary classifier (Odena et al., 2017) or the mutual information loss of (Chen et al., 2016)). A more theoretical analysis of convergence and equilibrium existence in the loss landscape motivated a multiple-generator, multiple-discriminator mixture in (Arora et al., 2017). We discuss in detail the works with the most resemblance to our approach here:

**DeLiGAN** The DeLiGAN (Gurumurthy et al., 2017) was designed to handle diverse datasets with a limited amount of datapoints. It used a single generator and a Gaussian mixture model latent space. To train, a single random $Gaussian_i$ out of the mixture is chosen, and then they added $\mu_i$ to the $Normal(0, 1)$ noise and multiplied it by $\sigma_i$, with both $\mu_i$ and $\sigma_i$ as learnable parameters. This differs from our work because while the noise is separated into different components, the probability of selecting each component is cut off from the gradient information in the model and is not differentiable (each Gaussian is selected with an equal probability, and this never changes). Also, every component of the noise is parameterized as a Gaussian. Finally, only one component of the noise is trained at a time (a single $\mu_i$ and $\sigma_i$ is randomly selected for each training batch), while our model learns to model the data over the full collection of generators in each minibatch.

**MGAN** The MGAN (Hoang et al., 2018) focused on the problem of mode collapse and addressed it by using multiple generators which are really the same network except for the first linear projection layer. They introduced a new loss term into the traditional GAN training: to encourage the generators to learn different parts of the data space, a lower bound on mutual information between the generated images and the generator they came from was maximized. This is helpful because the generators share almost weights between them and otherwise may redundantly use multiple generators to cover the same part of the space. Unlike in our work, they use a single noise source and let the single first layer of the generators learn to project it to different parts of the space before going through the same convolutions. In our work, this transformation of the noise before going to each generator is done with a separate network which gets gradient information through the generator, but is not optimized jointly with the generator weights. Moreover, like the DeLiGAN, the probability over the multiple generators was assumed to be fixed and uniform.

**DMWGANPL** The DMWGANPL (Khayatkhoei et al., 2018) exclusively viewed multi-generator models as a solution for disconnected manifolds. Each generator is given the same single noise sample, and the same mutual information criteria (termed $Q(G_i|x)$, the probability that $x$ came from generator $G_i$) as the MGAN was used to ensure each generator learned a different part of the space. Unlike the previous works, they do not assume an equal probability of selecting each generator. Instead, they sample each generator $G_i$ with probability $r_i$. After each step of the generator and discriminator during training, the $r_i$'s are updated to maximize mutual information between their distribution and $Q(G_i|x)$. This has the primary effect of not sampling redundant generators whose output is hard to distinguish from another generator's output, and is completely disassociated from the minimax GAN game. Each generator gets the same noise sample that takes a single parametric form ($Normal(0, 1)$), and the effect this has on the minimax game and the quality of generated images is only indirect and tangential to the objective being minimized.

## 3 MODEL

Let $X \sim P_X, x_i \in \mathbb{R}^d, i = 1...N_x$ be a sample of $N_x$ points from a $d$-dimensional distribution $P_X$. We seek a generator $G$ that learns to mimic $P_X$ by mapping from a noise distribution $Z \sim P_Z$. To do this, we train a discriminator $D$ and pit them against each other in the standard GAN framework:

$$\min_G \max_D L_{GAN} = \mathbb{E}_{x \sim P_x}[log(D(x))] + \mathbb{E}_{z \sim P_z}[log(1 - D(G(z)))]$$

where $G$ tries to minimize and $D$ tries to maximize this objective.

Motivated by the success of ensemble models (Miller & Ehret, 2002; Wang & Yao, 2009; Hinton et al., 2015; Whitehead & Yaeger, 2010), our NPGAN represents the generating function $G$ with multiple distinct generators of the same architecture. However, rather than simply averaging equal, independent, randomly initializing models to take advantage of uncorrelated errors, we adopt insights from machine teaching (Mei & Zhu, 2015) and knowledge distillation (Hinton et al., 2015).

---

**Algorithm 1** Calculating the loss $L_{GAN}$ optimized during training.

---

$G$: Generator, $D$: Discriminator, $NP$: Noise-Prior Network
Sample $x_{k=1...n} \sim X$
Sample $z_{j=1...n} \sim N(0,1)$
**function** LOSS$(x,z)$
    $\pi_j^{i=1...k} \leftarrow NP(z_j)$                                           $\triangleright$ Probability point $j$ is sampled from generator $i$
    $D_{fake} \leftarrow \sum_{ij} \left( \pi_j^i \cdot D(G_i(z_j)) \right)$
    $D_{real} \leftarrow \sum_{k} D(x_k)$
    $L_{GAN} \leftarrow log(D_{real}) + log(1 - D_{fake})$                     $\triangleright$ Expression for G/D to minimize/maximize
    **return** $L_{GAN}$
**end function**

---

We use a teacher network to select a generator $G_i$ conditioned on the particular input it sees. By learning a prior over the generators conditioned on the noise, this Noise Prior (NP) network delegates each input point to the appropriate generator that is optimally prepared to handle it. Thus, our total generator $G$ can be decomposed into:

$$G(z) = NP(G_i|z) \cdot G_i(z)$$

When traditionally training a GAN, the generator and discriminator alternate gradient descent steps, allowing gradient information to flow through the other network while keeping it fixed and only optimizing with respect to the given network's parameters. We extend this to our third noise prior network $NP$, allowing gradient information to flow through the fixed generators $G_i$ and discriminator $D$ while optimizing the GAN loss with respect to the parameters of $NP$. During training, we let both the NP network and the generators use a soft version of the GAN loss, weighting each generator output by the learned probabilities $NP(G_i|z)$. Then, during inference, the choice of $G_i$ is sampled from this learned prior. The full details of the training procedure are given in Algorithm 1.

The $NP$ network looks at the sample from $Z$ and determines how best to divide it across the generators to achieve the goal of modeling $P_X$. In the special case where $P_X$ is disconnected, $NP$ can divide $Z$ in multiple ways such as giving each generator a continuous area of $Z$, or giving some generators multiple disconnected areas of $Z$ (as we demonstrate later in the experiments). Nothing in our model formulation is specifically designed to model disconnectedness or any other specific property in $P_X$, however, so $NP$ can learn to divide the sample from $Z$ in whatever way is most appropriate for the given shape of $P_X$.

Thus, we model a distribution over our generators rather than simply sampling them uniformly and concatenating their output. Moreover, we learn this distribution over the generators with another network which is conditioned on the input noise, allowing it to choose the shape of input each generator receives. This network does not optimize a separate loss that is a heuristic indirectly related to the GAN framework, but directly participates in the GAN minimax game. To summarize, we fully and flexibly incorporate the multiple-generator framework into a GAN such that the model can learn for itself how best to use the generators. This is achieved by modeling a prior over the generators that is conditioned on the noise input and optimizing it with respect to the GAN loss directly.

## 4 EXPERIMENTS

### 4.1 DISCONNECTED MANIFOLDS

Our first experimental dataset (Figure 2) consists of a mixture of samples from two-dimensional Gaussians such that the three Gaussians are not sampled with equal probability (7000, 5000, and 3000 points, respectively). We compare our NPGAN's ability to model this distribution to a single generator model, MGAN (Hoang et al., 2018), DMWGANPL (Khayatkhoei et al., 2018), and DeLiGAN (Gurumurthy et al., 2017). The noise distribution for each model was a 100-dimensional $Uniform(-1, 1)$ except the DeLiGAN, which requires samples from $Normal(0, 1)$. The generators in each case share the same architecture of three layers with 200-100-20 neurons per layer and Leaky ReLU activations. The discriminator in all cases had three layers of 1000-200-100 neurons and used minibatch discrimination (Salimans et al., 2016).

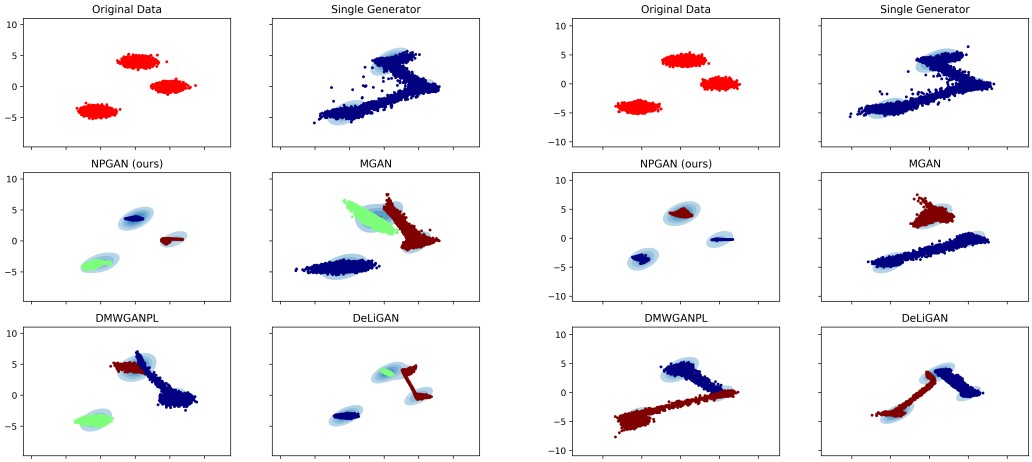

Figure 2: A single generator fails to capture the underlying structure without generating points off the support. Only our NPGAN learns to sample the generators in proportion to the data (the generator is indicated by the point's color).

Figure 3: Only our NPGAN is able to learn to create a discontinuity in the support of one generator and capture three manifolds with just two generators (the generator is indicated by the point's color).

Initially obvious is that a single generator cannot model this distribution without generating trailing points that connect the manifolds. By looking at the underlying density plots, we see that *most* of the generated data lies on the manifolds, in terms of proportions of points. However, when densely sampling the noise distribution, these few off-support outliers still arise. We then evaluate all of the multi-generator models with three generators, which we know to be the true underlying number of disconnected manifolds in this synthetic situation. The MGAN and DeLiGAN fail to model each manifold with a distinct generator and thus cover multiple manifolds with one of their generators and produce a trail of points in between. This failure stems from their sampling the generators with a fixed, equal probability. Since the disconnected manifolds do not have exactly the sample probability, their model formulations cannot effectively manage this situation. The DMWGANPL does learn a prior over the generators, but this prior only learned to eliminate redundant generators. Thus, it does learn an unequal sampling of the generators and each generator produces points that are distinct from the other generators, but does so without accurately modeling the data. The NPGAN, however, assigns each manifold to an individual generator and matches the data distribution without generating any off-manifold points. This is confirmed quantitatively in Table 1, where we measure the percentage of points each model generates that are off the manifold, which we define to be any point farther from the center of any Gaussian than the largest distance of any of the true points. There we see that the NPGAN generates no points off the manifold, while the other models are all forced to generate a trail of points connecting two of the Gaussians.

We next demonstrate the improved flexibility of the NPGAN over previous models by choosing two generators, imagining ourselves in the realistic case of not knowing the true underlying number of disconnected manifolds in a particular dataset (Figure 3). In this case, all of the other models must inevitably cover two manifolds with a single generator. Since each generator receives a continuous, unaltered noise distribution, this means they produces points off the manifold (Table 1). The NPGAN alone learns to model three disconnected manifolds with two generators without generating off-manifold points in between.

To investigate how the NPGAN achieves this, we learn another model with $Z \sim Uniform(-1, 1) \in \mathbb{R}^2$, so that we can plot the noise in two dimensions. In Figure 4a and 4c, we see the noise space for two and three generators, respectively. Notably, in both cases there are three partitions, no matter the number of generators. By learning to give one generator disconnected input, the $NP$ network effectively models a third manifold without having a dedicated generator responsible for it. Viewing the latent space also informs us how the NPGAN can easily model non-equally sampled manifolds,

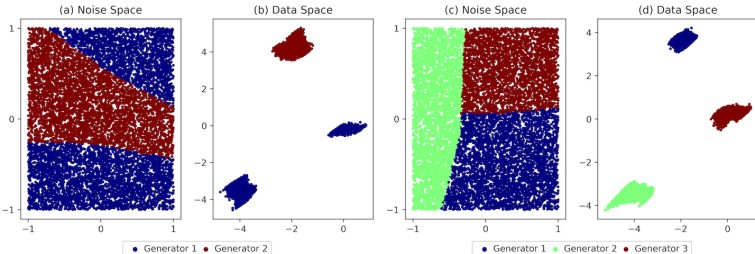

Figure 4: We investigate the NP network by using two-dimensional uniform noise and plotting the learned prior over the generators. The three unequally sampled Gaussians in the data can have their density matched with three generators (c-d) or by creating a discontinuity with two generators (a-b).

|  | 2Gen Gaussian | 3Gen Gaussian | 2Gen Parabolas | 3Gen Parabolas |
|---|---|---|---|---|
| NPGAN | **0.000** | **0.000** | **0.003** | **0.007** |
| MGAN | 0.031 | 0.162 | 0.088 | 0.083 |
| DMWGANPL | 0.037 | 0.026 | 0.076 | 0.082 |
| DeLiGAN | 0.134 | 0.055 | 0.196 | 0.320 |

Table 1: Scores for each model on each artificial dataset and the number of generators used (2Gen or 3Gen). For the Gaussian data, the score is the percentage of generated points off the manifold. For the parabolas, the score represents the percentage of real points without any generated point in its neighborhood.

as well, as the size of each partition of the noise space expands or contracts to match the underlying data distribution.

## 4.2 COMPLEX CONNECTED MANIFOLDS

Our next dataset explores the case where the underlying data distribution is complex but not necessarily disconnected. The other models have design choices to specifically target distinct areas of the data space with each generator. While single generator networks have difficulty with disconnected parts of the data space, there are many other ways the data distribution can be difficult for a single generator to model that have nothing to do with disconnectedness. Since the NPGAN gives the $NP$ network full flexibility to shape the input for each generator however it needs to in order to beat the discriminator, it can aid in generating complex shapes of any kind.

To investigate this we create a distribution of intersecting parabolas and test it with two and three generators for each model (Figures 5 and 6). As before, this complex shape is too difficult for a single generator to effectively model. In the DeLiGAN, the equal probability Gaussians trained alternatingly are unable to coordinate with each other and capture any of the tails of the two parabolas. The MGAN and DMWGANPL have the mutual information penalty that pushes the generated points for different generators away from each other. This not only keeps them from learning to generate intersecting shapes, but it pushes the optimization away from any solution where it requires a complex function to know which generator a particular point came from. The NPGAN, on the other hand, effectively models the data distribution with just two generators while finding a different but equally effective way of modeling it with three generators. As opposed to the previous Gaussian example, the problem here is not generating points off the manifold but leaving parts of the true manifold unmodeled. Thus, to quantitatively evaluate this dataset, we calculate the percentage of real points that do not have a generated point within an $\epsilon$-ball centered on it ($\epsilon = .001$). Table 1 confirms that the other models leave significant parts of the tails unmodeled, representing as much as 32% of the data in the most extreme case. The NPGAN's low score with both two and three generators corroborates that it can not only help in modeling complex distributions, but that the flexible formulation makes it robust to the specific number of generators chosen.

## 4.3 CELEBA+PHOTO

To test the NPGAN's ability to model disconnected, unevenly sampled manifolds on real images, we combine two distinct datasets. We take 1500 images randomly from the CelebA (Liu et al., 2015) dataset and combine them with the 6000 photographs dataset from (Zhu et al., 2017). To

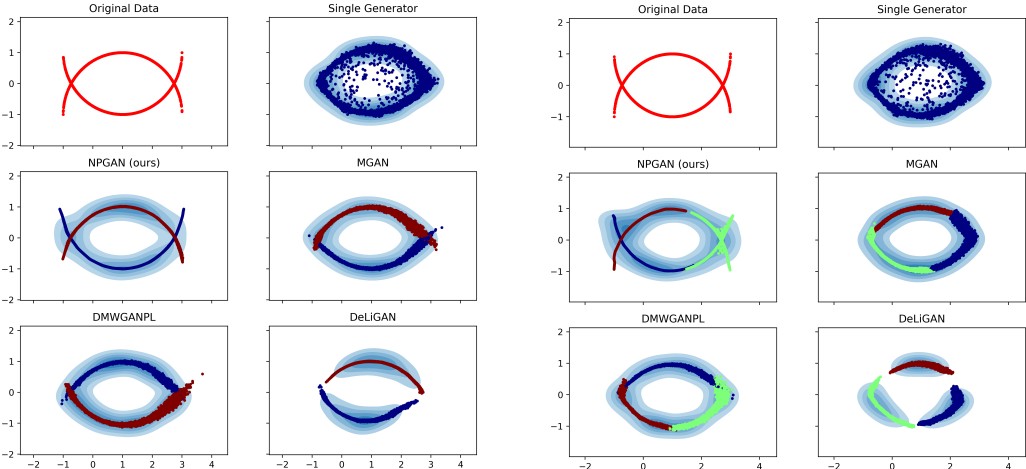

Figure 5: A single generator is unable to model this non-disconnected data. The disconnected assumption of the other models forces each generator to produce points that are separable from other generators'.

Figure 6: The parabola dataset is naturally solved with two generators, but the NPGAN is robust to the number of generators that are chosen. The other models can only work if the precise optimal number of generators is known *a priori*.

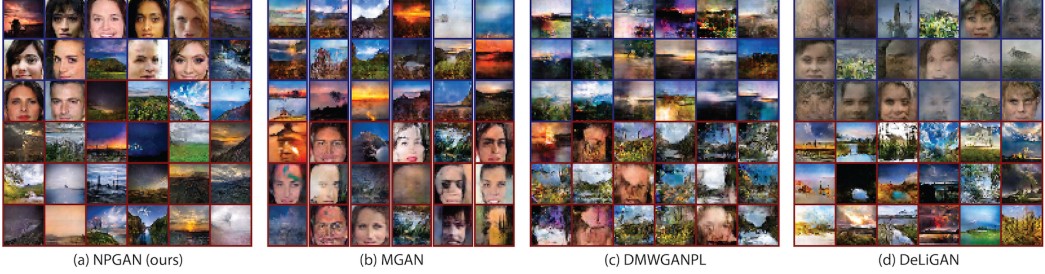

(a) NPGAN (ours)  (b) MGAN  (c) DMWGANPL  (d) DeLiGAN

Figure 7: Randomly selected images from each model trained on the CelebA+Photo dataset.

effectively match this data distribution with two generators, the models will have to either learn to sample generators at a differential rate, or have one generator cover a discontinuity (or both).

The images were resized to 32x32, and all models use a DCGAN architecture (Radford et al., 2015), with three convolutional transpose layers in the generators and three convolutional layers in the discriminator. Each convolution used stride length two, kernel size three, batch normalization on all layers except the first layer of the discriminator, and ReLU activations in the generator with Leaky ReLU activations in the discriminator. Training was performed on minibatches of 32 with an Adam optimizer (Kingma & Ba, 2014) with learning rate 0.0001. In the MGAN and DeLiGAN, the generators are all the same network except for the initial linear projection of the noise (or the adding of the generator-specific mean and the multiplying of the generator-specific standard deviation in the DeLiGAN). In our NPGAN and the DMWGANPL, the generators do not share weights. To compensate for the increased capacity that this would otherwise provide, we decrease the number of filters per generator learned to keep the total number of parameters across all models (within $1\%$).

Figure 7 shows randomly selected images from each model, and there we can see the consequences of the MGAN and DeLiGAN sampling generators at a fixed rate and giving each generator the same continuous noise. In each case, one of the generators effectively generates photos, but the other generator gets caught in between generating photos and CelebA images, producing many blurry images. The DMWGANPL samples generators at a different rate, but again did so ineffectively: one generator makes photos, but the other generator makes both CelebA images and some photos that are not being made by the other generator. Even though it is an imperfect measure of capturing outliers, the FID scores reported in Table 2 show that the imbalance affects their ability to model the

underlying dataset, too. To add an uncertainty measure to this score, we average the last three model checkpoints and report the mean and standard deviation. In Figure 7, we see the NPGAN learns to sample more from the generator that exclusively makes photos, while also using its ability to create discontinuity in its input to allow the other generator to make both CelebA images and a few realistic photos. Only the NPGAN is able to effectively model this disconnected, unevenly sampled dataset.

## 4.4 COMPLEX-BUT-CONNECTED IMAGE DATASET

In this section, we explore how connectedness affects the results of the models for image datasets. The previous works on multiple generators have emphasized disconnectedness, but we show here that the NPGAN outperforms the alternatives even without disconnected data. The effects of other properties, like class/mode imbalance, dominate the results. We test this notion by modifying the dataset from the previ-

| FID Score | Face-Photo | ConnectedFace-Photo |
|---|---|---|
| NPGAN | 58.9 +/- 2.4 | 51.7 +/- 4.1 |
| MGAN | 65.5 +/- 1.5 | 63.1 +/- 3.4 |
| DeLiGAN | 64.1 +/- 3.1 | 68.6 +/- 2.6 |
| DMWGANPL | 81.0 +/- 6.2 | 83.4 +/- 5.5 |
| WGAN-GP | 66.5 +/- 4.9 | 69.3 +/- 5.9 |

Table 2: FID Scores for all models.

ous section to create a connection between the Face and Photo images. To do this, we randomly choose images in each dataset and perform linear interpolation between them with a mixing coefficient $\alpha$ chosen from a $Uniform(0, 1)$ distribution. We add these interpolations to the Face+Photo dataset to make a ConnectedFace+Photo dataset. Conceptually, ConnectedFace+Photo takes the shape of a "barbell" with a narrow trail connecting two areas of density in data space.

We then repeat the experiment of the previous section and report the results. Notably, the quantitative results remain essentially the same. This can be explained with a couple of observations. First, as in the artificial cases, the other models have difficulty dealing with density imbalances, and this difficulty dominates the effects of whether the data is disconnected or not. Second, as previously discussed, the FID scores in Table 2 are affected most strongly by model performance where most of the data density is as opposed to a few bad trailing outlier points. Nonetheless, the presence of wrong off-manifold outliers like those in Figure 7b-d could be severely problematic in contexts with a higher sensitivity to outliers than the FID score captures.

## 4.5 CIFAR

Next, we explore the NPGAN's ability to model the canonical CIFAR10 dataset (Krizhevsky et al., 2014). Unlike in the previous case where the disconnectedness in the data was drastic enough to measurably affect sample quality as measured by FID, here our NPGAN produced essentially the same FID as our code with one-generator (26.4 to 25.8). However, as previously discussed, FID is not a good measure of whether a model produced outliers or not, since generating 1% bad samples off the manifold will be unnoticed in FID score if coupled with a slight

| | FaceBed | CIFAR |
|---|---|---|
| NPGAN | **16.39** | **18.94** |
| WGANGP | 17.84 | 20.74 |
| DMWGANPL | 17.51 | 19.69 |
| DeLiGAN | 17.69 | 19.61 |
| MGAN | 17.43 | 19.58 |

Table 3: Outlier manifold distances for all models on FaceBed and CIFAR.

improvement of sample quality on the other 99% of the samples. With that in mind, we introduce a new measure of how bad a model's worst samples are: outlier manifold distance. Unlike FID, our outlier manifold distance is sensitive to a model generating outliers, irrespective of how good its best samples are. We calculate this distance by finding the average distance of the 1% furthest generated points from the real data manifold, as measured by the distance to the closest real point in the last feature map of the Inception network for each generated point. The outlier manifold distance for each model is then the average of the 1% largest distances (the 1% "most anomalous" points). In Table 3, we see that NPGAN has the best outlier manifold distance of all models. As a sanity check, we also calculate it on the previous FaceBed data, and show that it confirm quantitatively what we saw qualitatively and with FID score, that other models produce outliers that are worse than NG-PAN's worst samples. For space reasons, a more comprehensive investigation into the NPGAN's use of multiple generators on CIFAR we defer to the supplement.

## 5 DISCUSSION

We introduced a novel formulation of multiple-generator models with a prior over the generators, conditioned on the noise input. This results in improved expressivity and flexibility by shaping each generator's input specifically to best perform that generator's task.

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

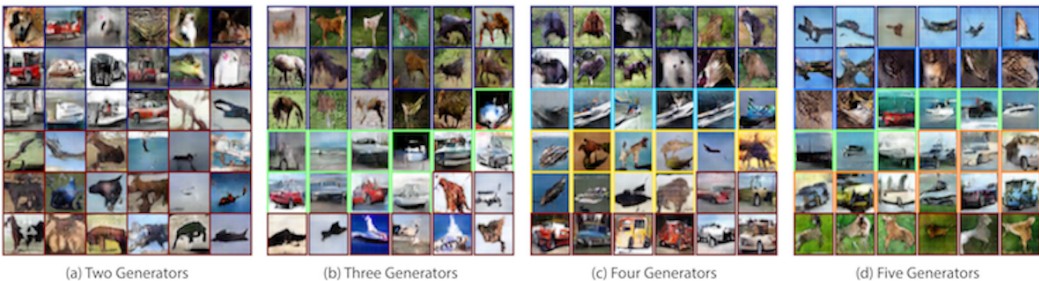

(a) Two Generators    (b) Three Generators    (c) Four Generators    (d) Five Generators

Figure A.1: NPGAN output, with the generator of the image represented by the color of the box, on the CIFAR dataset.

# A    SUPPLEMENT

## A.1    CIFAR

In this section, we elaborate on the CIFAR experiment from the main text. We use a more complicated architecture here with spectral normalization, self-attention, and ResNet connections, per the best achieving models to-date. We experimented using two, three, four, and five generators in the NPGAN architecture. Figure A.1 shows images generated by the NPGAN with each number of generators. With just two generators, each one creates a wide diversity of images. On the other hand, when increasing the number of generators, each one more homogeneous. For example, in the two generator model, one of them creates dogs, cars, and frogs, while in the five-generator model each generator has specialized to just birds in the sky or just cars.

Qualitatively, the noise prior is obviously learning a sensible split of the data across generators and each generator is outputting quality images. However, when comparing the two-generator, three-generator, four-generator, and five-generator versions of NPGAN to the baseline one-generator of the same model, we do not observe any improvement in FID score. This is unsurprising for the reasons mentioned in the main text. The FID scores treat all points equally across a generated dataset, and thus will be most strongly influenced by where the most points are. A relatively small number of outliers barely register by this metric.

Even current state-of-the-art image generation on CIFAR10 is no where close to perfectly modeling the data. When GANs are able to perfectly model the dataset except for trailing outliers between modes, we expect the NPGAN's improvements to be visible in FID scores on this dataset. Until then, the detection of a few bad outliers needs to be done with other evaluation techniques on this dataset.

With this caveat, we note that we could achieve an FID score of 26.4 with our NPGAN, compared to 25.8 with our code and one generator, which demonstrates that the NPGAN can scale to state-of-the-art architecture without suffering in quality. The NPGAN is robust to a connected dataset while simultaneously being able to automatically solve the problems of a disconnected dataset. Furthermore, this motivated the creation of our new outlier manifold distance metric, designed to be more sensitive to the creation of outliers than the FID score. Using this metric, we see NPGAN outperform all other models.

**Relation to Machine Teaching**    In (Zhu, 2013), an analogous question is posed: if a teacher network knows the function its student network is supposed to learn, what are the optimal training points to teach it as efficiently as possible? For students following a Bayesian learning approach, this is thought of as finding the best data points $D$ to make the desired model $\theta^*$, or minimizing with respect to $D$: $-log(p(\theta^*|D))$ . In our framework, the teacher network NP does not know the function its students should learn ahead-of-time, because this target is changing continually as the discriminator improves simultaneously. Nevertheless, the NP network is still learning to form the optimal curriculum for each individual student such that the collection of students best models the target function given the current parameters of the discriminator.

**Relation to knowledge distillation**    Our NP network also has links to the field of knowledge distillation (Kim & Rush, 2016; Chen et al., 2017; Furlanello et al., 2018; Wang et al., 2018), where a teacher network is trying to compress or distill the knowledge it has about a particular distribution into one or several (Hinton et al., 2015) smaller models. In the case of multiple smaller models, the teacher can be thought of as a generalist whose job it is to find the right specialist for a specific problem.

