# OpenReview forum: "Improving Multi-Manifold GANs with a Learned Noise Prior"
_ICLR.cc/2020/Conference — Reject_

### Official Review · AnonReviewer3 · 2019-10-07
**Official Blind Review #3**

**Rating:** 3

**Review:**

Overall:
IIUC, the main contribution of this paper is to take previous work on training GANs with
multiple generators and add a learned assignment of z -> G_i, so that we have
G(z) = \sum_i NP(G_i | z) * G_i(z)
where NP is this learned assignment.

I think that technically this idea is new,
but there is important related work [1] that (IMO)
  - does essentially the same thing
  - better experimentally validates the thing they do
  - was published in last ICLR.

I also have a number of issues with the experimental design: see my detailed comments below.
Finally, the technique seems like it can't easily scale up to larger GANs or data sets
because it requires instantiating many copies of the generator in memory?

For these reasons, I lean somewhat strongly toward rejection.


Detailed comments on draft:

It's worth noting that conditional GANs also sample from multiple disconnected `manifolds'.
I guess the value of adding your technique is that it can work without labels?

> Our approach differs significantly from those previously
strange sentence.

>  By dividing up the space into four slices...
I don't get this part.
If you fed all the slices to a single generator you could
generate four `disconnected manifolds' with just 1 generator, no?

> , multiple-discriminator mixture in (Arora et al., 2017)
Nit: you don't need the parens there, IMO.

> our model learns to model the data over the full collection of generators in each minibatch
Given that modern GAN techniques (e.g. bigGAN), this is going to have some pretty unpleasant
performance characteristics, right?
Each generator has to be instantiated in memory all the time.

> LGAN
Nit: surely, given the state of the GAN literature, this name has been used before.


> We compare our NPGAN’s ability to model this distribution to
The following baseline seems like it would be much simpler and get the job done:
Train a GAN on a prior that's a mixture of like 100 gaussians or whatever.
Then it seems like it could learn to assign 70, 50, and 30 of those modes
to each one of the gaussians in your underlying data?
I guess I haven't tried this myself, but it seems like a more fair comparison.
The way the current experiment is set up, you know a priori that your model
is the only one that can work.

Overall, I don't really understand why it's necessary to have multiple generators.
If I want the prior my generator `sees' to be disconnected, can't I just
pass samples from the prior through a standard relu network?
Surely a relu network can learn to separate one mode into two, and so forth.
But once you do that, you've essentially got [1].

> We next demonstrate
Would be nice to have a new subesection here.

I think your single-generator baseline in Fig 2 and 3 is unrealistically bad.
See fig 3 of [2], in which it looks bad but not nearly as bad as you've shown it.
I think this points to another issue with the experimental design.
If the generator is a big enough relu network, it ought to be able to automatically
generate `disconnected bits', but it has to be big enough, and you've compared a
generator with N parameters to 2 and 3 generators with N parameters each.
Moreover, your learned noise prior has some extra number of parameters (I don't see
where you described these but I may be missing it).

> A single generator is unable to model this non-disconnected data.
Again, I have a feeling that this is because of the small generator you use.
I'm pretty sure that a motivated person could get a normal GAN to model the
distribution in fig 5 reasonably well.

> The other models can only work if the precise optimal number of generators is known a
priori.
Isn't this not true for the method that throws out redundant generators?

The experiments described in 4.3 and 4.4 are pretty contrived.
It's ok to have some contrived experiments, but it seems like all experiments
on which you were able to provide evidence that your technique was helpful are contrived.
It also seems like the baselines you used are not obviously the right choice for these
experiments?

Re: your CIFAR experiments:
I think this supports my earlier claim that the extra parameters and bells and whistles
in modern GAN techniques *already implicitly do what your method is proposing to do*.
I think a lot of people miss this point when writing papers about GANs.
When people request comparisons against more modern techniques,
they're not (merely) being difficult:
they want to know if the technique you propose "stacks" with other techniques,
in the sense that your method is doing something that wasn't already being done implicitly
by the old methods.

References:
[1] On Self Modulation for Generative Adversarial Networks (https://arxiv.org/abs/1810.01365)
[2] Discriminator Rejection Sampling (https://arxiv.org/pdf/1810.06758.pdf)


**Experience Assessment:**

I have published in this field for several years.

**Review Assessment: Checking Correctness Of Derivations And Theory:**

N/A

**Review Assessment: Checking Correctness Of Experiments:**

I assessed the sensibility of the experiments.

**Review Assessment: Thoroughness In Paper Reading:**

I read the paper at least twice and used my best judgement in assessing the paper.

---

> ### Author Response · Authors · 2019-11-13
> **Response to Reviewer#3**
>
> Reviewer#3, we thank you for your time and effort in reading our manuscript!
>
> We respectfully disagree with this reviewer and believe that she/he misunderstood our goals. Succinctly stated our motivating problem is as follows: Since a generator learns a smooth function of its input (which is normally a smooth isotropic noise distribution), if the target distribution is disconnected, the generator necessarily will produce outlier samples in the “no-man’s land” in between. The single-generator model in Figure 2 is not “unrealistically bad”, because note the KDE contours underneath the scatter points. *Most* of the generated density is in the appropriate areas of the space. It has minimized the number of “trails of bread crumbs” it leaves between the modes, but it is absolutely unable to avoid producing some of these due to the continuous latent space that it is attempting to transform into disconnected manifolds. This motivates the whole line of work, that we believe we meaningfully move forward, by using a multi-generator model.
>
> We also believe that the “Self-Modulation” work is extremely different to our work [1]. Not only is it a different technique, but it does not even attempt to address the same general problem area that ours (or related work like MGAN and DMWGANPL) addresses, namely disconnected data. By simply performing another transformation on each *internal* feature of the network (i.e. multiplying by alpha and adding beta, two learnable parameters), their technique does not modify the latent space at all, nor manipulate it differently in any way. Nor does it use multiple generators. Most importantly, the output from their network will still be smooth, and thus face the same problem in producing outliers between disconnected regions of the data.
>
> We also wish to correct an inaccurate belief about the scalability of our model: as mentioned, we reduce the number of parameters per generator such that the total number of parameters in our entire model is the same as in the baseline. So it is not comparing one generator with N parameters to k generators with N parameters (plus the small NP-net). Instead, we shrink our model so that the total number of parameters (NP-net included) match the baseline. Thus, fitting all of the generators into memory is no more expensive than fitting the single, larger, baseline generator into memory.
>
> For the alternative models that are suggested:
> — conditional GANs: These might be effective, but would require being able to identify and label all of the points in each disconnected mode of the data ahead of time, which our method does not require.
> — hardcoded n gaussians: This is actually implemented in work we compare to, DeLiGAN. These strong, fixed assumptions on the prior lead to poor performance, as demonstrated in our experiments.
> — use of ReLU: A ReLU cannot create discontinuity, as it is a continuous function. The *gradient* is discontinuous, but the function itself is not.
>
> Responses to the smaller points:
> — The example of slicing the latent noise into four contiguous slices: s1, s2, s3, s4. The key thing is that if they all get fed to the same generator, *they aren’t discontinuous anymore*. Only since they go to different generators would s1 and s3 be discontinuous for the first generator and s2 and s4 be discontinuous for the other generator (see Figure 4, left, for an illustration of this with 3 slices).
>
> — We argue the Face-Landscape experiment is not contrived, or at the very least, not contrived by us. This was introduced by the accepted ICLR paper on multiple-generators that we improve on, MGAN. We simply modified the experiment slightly to make the two manifolds not exactly 50%-50% split in the training data. Our baselines are appropriate for this experiment, as they introduce it in the first place, although we remove the crutch of guaranteeing the data is exactly evenly split.
>
> — We appreciate the suggestions on revisions to the wording and outlining of the paper to improve clarity and have incorporated them in the new version!
>
> Overall, we hope you reconsider your review in light of the discussion on the problem of modeling discontinuities with a GAN. The problems of using a single smooth, continuous function are often ignored, either by plotting density contours or by simply looking for the presence of “good” images from the generator. We believe the problem is important, even if subtle, and think that our work represents a step forward from the previously published work on this topic.
>
> [1] On Self Modulation for Generative Adversarial Networks (https://arxiv.org/abs/1810.01365)

---

### Official Review · AnonReviewer2 · 2019-10-08
**Official Blind Review #2**

**Rating:** 8

**Review:**

Comments:

-As I understand it this technique is relatively simple.  There are K distinct generators with different parameters.  There is a network NP(z) which maps from an input z to a weighting over the generators.  All of these generators are used during training and the loss on the discriminator is reweighted using NP(z).

-This doesn't involve any explicit discrete decisions so the entire thing along with NP(z) can be trained end-to-end using the usual backprop.

-It would be interesting to consider simply doing importance sampling after sampling NP(z) to select the most relevant generators to train on, as NP(z) is presumably much cheaper than G(z).

-One thing that I'd like to see is a variant where the "different" generators share almost all of their parameters, but have different batch-norm "mean/sigma" parameters.  This would be the same as conditional batch norm.  I think it would be interesting if this could have some of the strengths of this method while still allowing the generators to share most parameters.

Review:

This paper presents a simple yet well motivated new method for helping GANs to model disconnected manifolds.  There are a few more explorations that I'd like to see (discussed in comments), but I still appreciate this paper for directly addressing an important challenge.  The 2D toy experiments do a good job of illustrating why the method helps and also the improvements on the "disconnected face/landscape" dataset are quite good.  It is a bit disappointing that it doesn't help on CIFAR10, although I think it's reasonable to leave this for future work.

**Experience Assessment:**

I have published one or two papers in this area.

**Review Assessment: Checking Correctness Of Derivations And Theory:**

I assessed the sensibility of the derivations and theory.

**Review Assessment: Checking Correctness Of Experiments:**

I assessed the sensibility of the experiments.

**Review Assessment: Thoroughness In Paper Reading:**

N/A

---

> ### Author Response · Authors · 2019-11-13
> **Response to Reviewer#2**
>
> Reviewer#2, thank you for such a strong recommendation of acceptance, and your overall time and effort spent in reading our manuscript!
>
> The variant of having the generators differ only in batch-norm parameters is very interesting! In our experiments, we keep the total number of parameters in the model the same by shrinking the size of each generator (for fair comparison to the baselines). This would you suggest be a way of implementing the core NP(z) idea essentially without increasing the number of parameters at all (save for the handful of batch-norm parameters and the very small NP-net itself).
>
> Your idea of using NP(z) to do importance sampling is fascinating! You are indeed correct, NP(z) is very cheap to compute. If we understand correctly, this importance sampling would allow the network to dynamically train more on generators with lower quality, and save on resources training generators currently of sufficient quality for that given point in training. One potential drawback to this method is that when filtering some generator output, it has the potential of making the gradients not point towards exact density matching (i.e. if D does not receive samples in a given region from a high-quality generator than the lower-quality generator may move towards generating more samples in that region to compensate, beyond just making its quality higher). We would love to explore this further in future work!
>
> Lastly, we agree that the most important results we have are about the off-manifold demonstrations when we know we have disconnected data. As for our CIFAR results, please see our newest draft (and our response to Reviewer#1) for the new metric we’ve introduced to attempt to measure this. FID measures something very different from the metrics used in our other experiments (overall sample quality on average, not how bad the worst samples are). We show NPGAN does indeed improve this aspect of sample quality. In the end, as you conclude, furthering this improvement and developing even better metrics for this would be an important direction for future work.

---

### Official Review · AnonReviewer1 · 2019-10-24
**Official Blind Review #1**

**Rating:** 6

**Review:**

This paper proposes to improve GAN by learning a function that splits the latent space in several parts and then feed each part to a different generator, this enables GAN to model distribution with disconnected support. They try the proposed approach on several toy examples where the support of the distribution are disconnected, the data is imbalanced or the support of the distribution intersect,  they in all this cases that the proposed approach improve performance over baseline, in particular the proposed method doesn't produce outliers. They also show how the method is robust to the choice of number of generators used. Finally they show improved performance on more challenging dataset in particular on the CelebA+Photo dataset as measured by FID score.

I'm slightly in favour to accept the paper. I think the idea is well motivated and shows real advantage over other methods on the toy experiments. The major downside of the paper is that the proposed method doesn't seem to improve that much in more realistic setting.

Main Argument:
+ The idea is well motivated and the paper precisely explain that they try to address the problem of modelling data when the manifold is disconnected and the class are imbalanced. They illustrate how the proposed approach is able to address this problem on some toy example

+ The paper also show how the method is robust to the choice of number of generators

+ Figure 7 and A1 are quite interesting showing how the different generators can learn different part of the data like different class.

- The main counterpoint is that the method doesn't seem to improve the performance that much in more realistic settings. The author point to the fact that this might be due to the fact that the metrics we used are not sensible to outliers. I wish the author had used another metric to show and confirm this hypothesis.

- I found the explanation of the proposed algorithm a bit confusing, it would be nicer if the final loss was clearly defined in the paper and the derivation of the loss explained. In particular in Algorithm 1 I don't understand why $L_{GAN} = D_{real} + (1-D_{fake})$ shouldn't there be some logarithm somewhere ?

- The author propose some measure of standard deviation for the different models but this computed on the three last model checkpoints. It would be much more valuable to compute the standard deviation and the mean with different seeds.

Minor comments:
- Conflicting notation: In the related work you use $G_i$ to denote a random Gaussian, the same notation is used to denote the generator.

- The part about Machine Teaching and knowledge distillation seem a bit irrelevant to me. I don't really understand what they bring to the paper. Also this would save some space and enable to put the CIFAR10 results in the paper

- In the toy experiments I would be curious to see the results when the different modes have the same probability.

- It would be interesting to have the influence of the number of generators on the FID.

**Experience Assessment:**

I have published in this field for several years.

**Review Assessment: Checking Correctness Of Derivations And Theory:**

N/A

**Review Assessment: Checking Correctness Of Experiments:**

I carefully checked the experiments.

**Review Assessment: Thoroughness In Paper Reading:**

I read the paper at least twice and used my best judgement in assessing the paper.

---

> ### Author Response · Authors · 2019-11-13
> **Response to Reviewer#1**
>
> Reviewer#1, we are glad you recommend our paper for acceptance and first wish to thank you for your time and effort in reading it!
>
> We agree with the strengths that you identify, and we also agree with several of your points about where there’s room for improvement. In our revision, we have tried to clarify the final loss, and made the algorithm more clear. We have also removed the notation of G referring to a Gaussian in the related work section, where we took the notation from the previous work itself without considering that it clashed with our notation. And if you feel our connection to Machine Teaching and Knowledge Distillation not to be worthwhile, we would happily switch these into the supplement if you prefer (which we have done in the revision). Also, with more time, we will definitely do as you suggest and rerun with different initializations to measure the variation across runs for all models and datasets. Thank you for all of your help in these improvements!
>
> We would, however, like to respectfully push back on the critique of it “not improving the performance that much in more realistic settings”. The NPGAN actually does improve performance in realistic settings. However, the issue is that the standard GAN performance measure FID score does not penalize off-manifold outliers very much. It is perhaps not a surprise that this is the case, because it is very hard to detect and quantify being “off the manifold” for an arbitrary image dataset. Despite this, we have made an attempt to do so in our revision by adding what we call an “outlier manifold distance” score. As opposed to FID (where generating 1% bad samples off the manifold will not affect the FID score if coupled with a slight improvement of sample quality on the other 99%), the outlier manifold distance is sensitive to how bad a model’s worst samples are.  We calculate this distance by finding the average distance of the 1% furthest generated points from the real data manifold, using the representations in the last feature layer of the Inception network as the data manifold. For more details, please see the new version of the paper. Albeit imperfect, this measure shows our NPGAN producing points that are closer to the real data than the other models. Coupled with the FID score as an overall measure of sample quality that shows the NPGAN does not deteriorate the quality of the best samples on CIFAR, we believe this offers a more complete measure of the model’s performance on.
>
> In practice, for a network to be deployed in particular sensitive operations, we don’t believe it is unreasonable to ask for a generative model to prioritize avoiding outliers over producing an optimal FID score. The latter is, after all, merely a heuristic that offers one particular way of aggregating an “overall” quality of samples. We simply argue that the community should work on developing GANs that can avoid generating off-manifold outliers in addition to the plethora of work on standard FID-optimizing varieties.
>
> Lastly, on the toy experiments with equal probability, we would expect all of the models to succeed at this task (as it fits the assumptions of the other models), and is very similar to experiments run in their papers already. As we saw little new knowledge to be gained from such an experiment, we decided to only go for the more informative, novel version with unbalanced modes.

---

### Decision · Program_Chairs · 2019-12-19

**Decision:**

Reject

**Comment:**

This paper introduces a modified GAN architecture that looks a lot like a mixture of experts, to address the problem of learning multiple disconnected manifolds.  They show this method helps on 2D toy experiments, and artificial tasks where different datasets are combined, but not on CIFAR.  They also introduced a new variant of FID that they claim is more sensitive to the improvements made by their model.

R2 didn't seem to think too hard about the paper, and R3 seemed a bit dismissive.

Overall the idea seems sensible but the particulars of this approach aren't all that well-motivated in my opinion, especially since the cost of the generator is increased.  Why not just use a mixture of Gaussians in the original untransformed space?

I also found the toy experiments unconvincing, particularly the claim that a standard GAN couldn't learn a mixture of 3 Gaussians.  Learning a mixture of 8 Gaussians was one of the results in the unrolled GAN paper, for instance.

The results on the mixed datasets experiments seem encouraging, but I'm afraid that proposing a new GAN architecture in 2019 requires even more baselines than the authors compared against, and the fact that the task was artificially constructed undercuts its importance.